# Interventions combining mindfulness training with non-invasive brain stimulation and their impact on mental health outcomes: Protocol for a systematic review and meta-analysis of randomized controlled trials

**Anastasia Demina[1,2]\*, Benjamin Petit[1], Vincent Meille[1], Florent Lebon[2], Benoit Trojak[1,2]**

**1** Addiction Medicine Department, Dijon Bourgogne University Hospital, Dijon, France, **2** INSERM U1093, CAPS, Université de Bourgogne, UFR STAPS, Dijon, France

\* anastasia.demina@chu-dijon.fr

## Abstract

### Background

Mindfulness training programs and non-invasive brain stimulation are both evidence-based interventions that have applications in mental health disorders. While both have showed promising results on a range of symptoms related to mental health, their combination has more recently grabbed the attention of researchers. There is a theoretical framework for their synergistic effects, and these effects can be tested through a variety of neurophysiological and clinical outcomes. This emerging field of research, which is regularly extended with new trials, has not yet been systematically reviewed. This systematic review protocol aims to present a rationale for combining these two interventions and to document the methodical approach to our systematic review before data extraction.

### Methods and analysis

Four electronic databases (Medline, EMBASE, CENTRAL, PsycINFO) and three clinical trial registries (Clinical Trials, EU Trials, WHO ICTRP) were searched. All randomized controlled trials testing the combination of mindfulness-based interventions and non-invasive brain stimulation in humans will be included. As primary outcome, data on change in anxiety and depression symptoms from baseline, and, as secondary outcomes, other mental health outcomes data will be gathered. Data will be extracted independently by two authors using a predefined extraction form. Depending on the clinical heterogeneity of the included studies, the research team will decide whether a quantitative synthesis is appropriate for each of the predefined outcomes. If there is considerable statistical heterogeneity, subgroup analyses and meta-regression will be performed. Bias will be assessed using a revised Cochrane risk-of-bias tool for randomized trials and the strength of evidence in our review will be assessed using the GRADE form in GRADEPro.

**Data Availability Statement:** Our systematic review and meta-analysis will gather already available data from the included trials.

**Funding:** The author(s) received no specific funding for this work.

**Competing interests:** The authors have declared that no competing interests exist.

**Abbreviations:** MBCT, Mindfulness Based Cognitive Therapy; MBI, Mindfulness Based Interventions; MBRP, Mindfulness Based Relapse Prevention; MBSR, Mindfulness Based Stress Reduction; MeSH, medical subject headings; NIBS, Non-invasive brain stimulation; RCT, randomized controlled trial; rTMS, repetitive transcranial magnetic stimulation; tDCS, transcranial direct current stimulation; WHO, the World Health Organization.

We started our scoping searches in November 2022. This systematic review and meta-analysis protocol was finished and submitted before the end of the independent full-text selection process by two members of the team.

## Ethics and dissemination

Ethics approval and consent to participate were not applicable to our systematic review. Our dissemination plan includes the publication of our systematic review and meta-analysis in an international peer-reviewed journal as well as international communication of our results.

## Trial registration

PROSPERO registration number CRD42022353971.

## Background

Mindfulness is commonly described as a modified state of consciousness in which a person deliberately observes their experience without passing judgment [1]. Generally, this state of consciousness is obtained through mindfulness meditation practice, and today it has multiple applications in medicine through evidence-based training programs such as Mindfulness Based Stress Reduction (MBSR), Mindfulness Based Cognitive Therapy (MBCT) and Mindfulness Base Relapse Prevention (MBRP) [2–4]. The common goal of all mindfulness-based programs is to cultivate a moment-to-moment observation of experiences, whether they are pleasant, neutral or unpleasant, and to distance oneself from emotions, thoughts and beliefs, observing those as mental constructs rather than letting them dictate life choices [5]. Mindfulness training programs or mindfulness-based interventions (MBI) have beneficial effects on stress, emotion regulation, and on a range of other mental health outcomes such as depression symptoms, anxiety symptoms, pain experience and sleep disturbances [6–8]. It has applications in psychiatry, cardiovascular disease, oncology, diabetes, renal failure and addiction medicine, to name a few [9].

While it appears to be beneficial in a wide variety of conditions, mindfulness meditation is also considered to be hard to learn. It requires multiple competences such as awareness of one's bodily sensations and breathing, capacity to notice mind-wandering during practice, and the capacity to observe thoughts and emotions without engaging in them. Despite the multi-week duration of classic mindfulness training programs, the learning curve is considered to be sharp [10]. Not only it could preclude adherence to these interventions and their generalization, but it could also limit the access to mindfulness meditation for those who have trouble with this complex training due to their disease burden. Therefore, it appears important to explore possible parameters enabling mindfulness training facilitation.

There is a growing scientific interest in combining different evidence-based treatments based on their possible synergistic effect [10]. In case of MBI, multiple recent studies suggest combining it with non-invasive brain stimulation (NIBS). NIBS refers to neuromodulation techniques acting on specifically predefined cortical areas, focally and on the deeper level [11]. Two of the most studied NIBS techniques are repetitive transcranial magnetic stimulation (rTMS) and transcranial direct current stimulation (tDCS) [12]. The former consists of delivering repetitive magnetic pulses through a metallic coil placed on the person's scalp, which generate brief electric currents in cerebral cortex. Depending on stimulation frequency, the effects of rTMS differ: low frequencies produce an inhibitory effect on neuronal activity while

higher frequencies produce an excitatory effect [13]. As for tDCS, it consists of delivering a weak current to predefined cortical areas with two saline-soaked surface sponge electrodes, resulting in the modulation of neuronal excitability [14]. These techniques are largely studied in psychiatry, pain management, neurology and addiction medicine, with demonstrated analgesic and antidepressant effect for high frequency rTMS, and promising yet sometimes divergent effects in substance use disorders and in cognitive function impairment [11,12,15].

Mindfulness meditation and NIBS combination is an emerging field of research. These two interventions are relatively simple to combine simultaneously or sequentially [10]. Both have established effects on outcomes of interest and relatively few adverse effects or counter-indications [5,11]. It is thus possible to study the complementary effect of their combination. In terms of theoretical framework, there is evidence that stimulation of cerebral regions that are already activated by a task results in enhancement of the stimulation effects [10]. Thus, NIBS applied to cortical areas activated by mindfulness meditation could be potentiated. Moreover, NIBS can provide promising procognitive effects, and it is a logical to hypothesize that it could facilitate mindfulness meditation training [10,15].

Considering the abundance of very recent literature exploring the effects of combined MBI and NIBS, qualitative and quantitative syntheses of available data are of interest. Thus, we designed a systematic review and meta-analysis to critically review all available and relevant literature on combined MBI and NIBS using predefined eligibility criteria.

Given the nature of both interventions, we can assume that there will be considerable variations between study protocols depending on the type of MBI and NIBS, the intensity and duration of the interventions, and the targeted cortical area. Consequentially, thorough and systematic exploration of existing protocols and their results is required to better understand how to combine these interventions on the bases of better efficacy, tolerance and feasibility parameters.

For this purpose, we will provide a synthesis of the different treatment combinations used in primary research. We will thoroughly explore participants' characteristics, their adherence parameters, and any reported adverse effects. If the extracted data on mental health outcomes is eligible for the meta-analysis, we will perform a quantitative synthesis for the interventions' effect size and direction exploration. We will also provide an exploration of heterogeneity in meta-analytical results. Given the emerging nature of this research, we aim to provide a summary of the reported and hypothesized mechanisms underlying the potential effects of combined MBI and NIBS. We also aim to explore and discuss methodological issues in this combined treatment research, namely blinding procedures, choice of control conditions, and sample sizes.

This systematic review and meta-analysis protocol aims to document our methodical approach in order to guarantee the trustworthiness of our review, in accordance with the international guidelines [16]. We also hypothesize that it could contribute to the reproducibility of our results.

We used PRISMA-P guidelines to ensure that all the recommended information is provided for this protocol [16], and we have preregistered our review with the International prospective register of systematic reviews with PROSPERO registration number CRD42022353971.

## Methods and analysis

### Eligibility criteria

**Study design and setting.** We will only include randomized controlled trials (RCTs) in humans. All other types of trials will be excluded from the review to avoid the bias of non-randomized and non-controlled studies. We will include studies both in clinical and laboratory

settings for exploratory purposes, while conducting separate analyses for each setting. Given our selection criteria, it is likely that most of the included studies were performed in tertiary care centers, which could potentially impact the generalizability of our findings.

**Participants.** We will include studies in adolescents, using the World Health Organization (WHO) definition of adolescence age limits as 10 to 19 years old, and adults [17].

Given the exploratory nature of our review, we chose to include both clinically diagnosed individuals with all types of medical conditions and healthy individuals. We will not pool both populations in the quantitative summary given the possibility for significant effect size differences between clinically diagnosed and healthy individuals.

**Interventions.** We will include trials in which MBI and NIBS are sequentially or simultaneously combined. MBI will be defined as interventions in which mindfulness competences are acquired through training program such as MBSR, MBCT, MBRP or other mindfulness training programs. In a separate analysis, we will explore studies with only a single mindfulness session at some point during the NIBS sessions. All types of NIBS will be included with all types of stimulation protocols.

**Comparator.** All types of control conditions will be accepted. Thus, comparators such as combinations of MBI and sham NIBS, sham MBI and sham NIBS, waiting list, and usual care will be accepted.

**Outcomes.** We chose to focus on all mental health outcomes because of the scope of the two interventions that are used to improve these outcomes [9,12].

We chose as primary outcomes:

- Changes from baseline in symptoms of depression measured by standardized scales following treatment.

- Changes from baseline in anxiety symptoms measured by standardized scales following treatment.

Symptoms of depression usually include sadness, anhedonia, guilt, sleep disturbance, suicidal thoughts, difficulty or incapacity to work and perform usual activities, weight fluctuations, anxiety, agitation, psychomotor retardation, concentration and memory difficulties, and they can also include a range of somatic symptoms such as muscle pain, gastro-intestinal disturbances including with lack of appetite and constipation, and loss of libido [18]. As for anxiety symptoms, they can consist of worrying, internal tension, fear, difficulty with concentration and memory, depressed mood, muscular symptoms such as pain and aches as well as sensory symptoms such as tinnitus and blurred vision, cardiovascular and respiratory symptoms such as palpitations, tachycardia and dyspnea, gastrointestinal symptoms with nausea, abdominal pain and lack of appetite, genitourinary symptoms with frequent micturition and loss of libido, and autonomic symptoms like dry mouth, flushing and sweating [19]. Only psychometrically validated standardized scales for rating depression and anxiety symptoms will be eligible for our analysis. We will summarize and define the psychometric properties of all the scales that were used in the selected trials.

As secondary outcomes, we will analyze the data related to stress, sleep quality, quality of life, problematic substance use, pain, and dispositional mindfulness. Stress is an umbrella term reflecting individual experience in a situation of imbalance between environmental demands and coping abilities [20]. In this review we will focus on the stress responses defined as emotional and cognitive reactions to stressors or stressful events, these reactions having an impact on an a person's health [21]. In fact, mindfulness training teaches individuals to disengage from stressful events thus decreasing emotional and cognitive reactivity to these events [2]. We will carefully explore the authors' definitions of stress and stress measures, and we will use all

psychometrically validated and appropriate scales in our analysis based on the resources provided by the Stress Measurement Network, University of California, San Francisco [22].

Another secondary outcome focuses on sleep quality. Sleep quality can be assessed using various sleep measures such as total sleep time, sleep maintenance, sleep efficiency, sleep onset latency, total wake time and disruptive events such as apnea. Sleep disturbances may occur in a variety of conditions. Because they alter quality of life and well-being, sleep quality is an important outcome for clinical research in mental health [23]. We will use validated scales in our analysis to appraise the effects of the interventions on sleep quality.

To determine the impact of the studied interventions on quality of life, generic and psychometrically validated scales of quality of life will be used. Quality of life measure is a vital measure of a general experience related to personal well-being [24]. Outcomes that can significantly alter quality of life, such as memory impairment and general cognitive impairment, will also be considered.

For pain evaluations, we will use both numerical assessments of intensity (visual analogue scale or other pain scales) and psychometrically validated scales or questionnaires about pain experience and interference. This aspect seems to be of particular interest in mindfulness training because mindfulness helps to cultivate observation and acceptance of painful experiences, potentially resulting in an altered experience of pain [25].

Dispositional mindfulness, also known as trait mindfulness, is related to a natural ability to relate to one's experience in a mindful way, with moment-to-moment awareness, engaging in a non-judgmental observation of stimuli, irrespective of one's experience with mindfulness meditation [26]. Mindfulness practice has been found to increase this trait, and a recent systematic review has shown that dispositional mindfulness and psychological health are positively related [27]. It is thus logical to investigate whether a combined intervention demonstrates the same effect. Dispositional mindfulness is measured using standardized scales and questionnaires.

We will explore data on adherence, adverse effects and acceptability to better report the benefit and harm of the studied interventions. Definitions of adherence, adverse effects and acceptability will be retrieved for each study, and quantitative data will be summarized if appropriate.

**Report characteristics.**    We will include studies in all languages, from all geographical areas, with no restrictions for year of publication. All types of RCT reports will be considered, including journal articles, conference abstracts, theses, book chapters, etc. If necessary, authors will be contacted to obtain supplementary information.

## Information sources

Four electronic databases (Medline, EMBASE, CENTRAL, PsycINFO) and three clinical trial registries (Clinical Trials, EU Trials, WHO ICTRP) were searched. To ensure literature saturation, we will hand search potentially relevant trials and screen the references in included studies. The first search took place in November 2022. We will update our search and include relevant data to our analysis toward the end of the review.

## Search strategy

We started our scoping searches in November 2022. Our literature search strategy used medical subject headings (MeSH), or equivalent when using Embase and PsycINFO, and text words related to the combination of MBI and NIBS. The Medline search strategy draft was created by a member of the research team with expertise in systematic review searches. This search strategy was pilot-tested and peer-reviewed by the research team members. No limits were

included in our search strategy. The draft for the Medline search strategy is available as a (S1 File). For other electronic databases and registries, our Medline search formula was adapted to the syntax and subject headings of the information source. We did not use filters in any of the information sources.

## Data management and study selection

We implemented all entries in a reference management software, Zotero, and, after automatic double removal, we implemented those in the systematic review automation tool Rayyan QCRI [28,29].

The study eligibility form was created by A.D. and peer-reviewed by all members of the research team. It is available as a (S1 Fig). Two raters (A.D. and B.T.) independently assessed titles and abstracts for inclusion criteria in blind mode. After unblinding, disagreements were resolved through discussion with a third rater (B.P.) The full texts of the included articles were then reimplemented on Rayyan QCRI for blind full-text assessment by two raters (A.D. and B.T.)

This systematic review and meta-analysis protocol was finished and submitted before the end of the independent full-text selection process by two members of the team. We are planning to finish full text selection process in March 2023. The conflicts will be resolved through discussion with the third rater (B.P.) We will provide an explanation for all excluded entries. Two raters (A.D. and B.T.) will also independently assess whether the entries should be included in the meta-analysis, and conflicts will be resolved through discussion with all members of the research team.

## Data collection process

The data extraction form was written by A.D. and peer-reviewed by all the members of the research team. The form includes the following items: study identification with first author's name and year of publication, study location, study design, population characteristics (sample size at baseline, gender, age, condition, experience in meditation), intervention characteristics (MBI type, MBI duration, NIBS type, intensity, duration, frequency, targeted areas, sequential or simultaneous administration), control characteristics, study timeline, outcome measures with timepoints. Data on pre-registered trial protocol availability, funding sources and potential conflict of interest will also be sought. Two authors will pilot-test this extraction form with 3 randomly selected trials. Then, we will adapt and apply the extraction form to all included publications. Data extraction will be independently performed by two authors (A.D. and B.T.). The published data will be used, and, in case of missing information, corresponding authors will be contacted through their institutional email address (maximum 3 email attempts within a 30-day period). In case of discrepancies in extracted data, two other members of the research team will adjudicate disagreements. Our extraction form is available as a (S2 and S3 Files).

In case of multiple reports of a single trial, they will be compared independently by two authors (A.D. and B.T.) using author names, intervention characteristics, study locations, date and duration of the studies, and sample sizes at baseline, as recommended by the Cochrane Handbook [30]. Only one of two identical trials will be included to avoid biases related to multiple inclusion of the same trial.

In case of crossover studies, only the data before crossover will be used. In case of more than one control condition, we will include only active control condition data. In case of different treatment intensities in multiple intervention arms, the results from the different arms will be combined and compared to the control condition.

## Risk of bias in the individual studies

Two authors (A.D. and B.T.) with experience in risk of bias evaluation will perform this assessment for each outcome in blind mode using a revised Cochrane risk-of-bias tool for randomized trials [31]. We will evaluate the following criteria: bias arising from randomization process, deviations from intended interventions, incomplete outcome data management, measure of the outcome, and selection of the reported result [31]. We will define an overall risk after categorizing each domain as "high risk", "low risk" or with "some concerns". If the blind independent evaluation does not match, we will obtain consensus through discussion with the third author (B.P.). Risk of bias evaluation will be visualized next to the forest plots for each outcome, as recommended in the Cochrane Handbook [30]. If the quantitative analysis is not performed, we will summarize the results using the Robvis visualization tool [32]. The risk of bias appraisal is a crucial part of evaluation of internal validity for the included studies. It will provide essential information to critically appraise the review's results in relation to the true effect of the interventions. Importantly, this evaluation promotes credibility and transparency through evaluation of individuals studies in comparison to their registered protocols. We will discuss the evaluation results for risk of bias and describe the implications for the validity of the review's overall findings.

## Data synthesis

We will conduct a meta-analytical synthesis if the extracted data is found to be appropriate for a quantitative synthesis. The final decision on the trials and outcomes to include in the meta-analysis will be obtained through discussion with all members of the research team. This decision will depend on our evaluation of clinical heterogeneity in the trials included in our review. Clinical heterogeneity of the included studies will be assessed through the clinical, methodological, and technical differences between the studies. Clinical and methodological differences will be appraised through analysis of the included population (age, sex ratio, education, diagnosis), percentage of patients lost to follow-up, and trial designs. Technical differences will be evaluated by comparing different intervention and control characteristics (types, technical features, duration). If significant clinical heterogeneity precludes a meta-analysis, we will provide a narrative synthesis describing the findings of our review using both tables and text formats. We will discuss both the results of individual trials and those of our review, as recommended by the Centre for Reviews and Dissemination [33].

For the narrative synthesis, we will first present our findings regarding the included population (sociodemographic information, diagnosis, experience of mindfulness meditation practice). Then we will describe the interventions (types of interventions and various ways to combine these). We will describe technical parameters of NIBS as well as the key features of the studied MBI. Thorough attention will be given to the conceptual differences between studies combining NIBS with evidence-based training programs and those that combine NIBS with only one mindfulness meditation session. We will then discuss the methodological features of the included trials, especially in terms of randomization process, blinding and choice of control condition.

We will present the outcome data starting with the findings on primary outcomes (symptoms of depression and anxiety). Then we will present synthetized data on secondary outcomes. To account for temporal evolution of hypothesized effects, our outcome data will be structured around different endpoints, from immediate post-treatment to multiple-month follow-up.

We will describe the risk of bias evaluation, confidence in cumulative evidence evaluation and will critically discuss the findings in light of these assessments.

If a quantitative synthesis is possible, the results will be summarized for each outcome as means and standard deviations. If standard errors are provided in the included studies, we will use the formula $SD = SE \times \sqrt{N}$ convert these to standard deviations [30]. To estimate the effect sizes for each outcome, the standardized mean difference will be used given the continuous nature of our predefined outcomes and possible variations between scales and measures used in the studies. We will implement the means, standard deviations and sample sizes for each outcome at post-treatment in intervention and control groups into Revman 5.4.1 software [34].

For quantitative synthesis, studies eligible for a pooled pairwise meta-analysis, outcomes and scales in case of use of multiple scales for a single outcome, will be selected through discussion with all members of the research team on the basis of clinical characteristics of the included trials.

Our analysis will include all measured post-treatment data. In case of missing data, we will contact the authors. If we are unable to gather the missing data, we will perform an imputation. We will also perform sensitivity analyses comparing per protocol and imputation models.

The inverse variance method and random effects model will be used to consider clinical et methodological differences between studies [35]. We will perform sensitivity analyses comparing the random effects model to the fixed effects model to evaluate the small sample bias. To estimate the effect size, we will use the Cohen's d [36]. The effect size is usually interpreted as small when d = 0.2, medium when d = 0.5, and large when d = 0.8 [37].

We will explore statistical heterogeneity of the studies with inconsistency measures, namely the Chi2 test with significance at <0.1 and the I2 statistic. The I2 statistic describes "the percentage of total variation across studies that is due to heterogeneity rather than chance" [38]. Higher values of I2, ranging from 0 to 100%, indicate higher heterogeneity across studies. If there is considerable heterogeneity between the studies (Chi2<0.1 and I2>50%), we will conduct subgroup analyses and meta-regression using participants' characteristics (age, sex ratio, education), type of NIBS (tDCS or rTMS or other), type of mindfulness training program, type of combination (simultaneous or sequential), treatment duration, proportion of patients lost to follow-up and high risk of bias as covariates.

## Meta-biases

The selective reporting biases will be assessed by comparing the pre-published protocols of the included trials to the published trials. We will compare the Methods and Results sections in published trials if the protocol is unavailable. If an adequate number of trials are included, we will explore the publication bias by analyzing the funnel plot. We will describe, interpret and critically discuss the results of each evaluation in the text of our review.

## Confidence in cumulative evidence

The strength of evidence in our review will be assessed using the GRADE form in GRADEPro [39]. The GRADE form will be completed by two authors (A.D. and B.T.) in blind mode. Consensus will be obtained through discussion with the third author (B.P.) in case of disagreements. We will use the Grade Handbook criteria to evaluate the confidence in cumulative evidence, namely risk of bias, inconsistency, indirectness, imprecision, publication bias, effect size, confounding factors and dose-effect gradient for each outcome [40].

## Supporting information

**S1 Checklist. PRISMA-P (Preferred Reporting Items for Systematic review and Meta-Analysis Protocols) 2015 checklist: Recommended items to address in a systematic review**

**protocol\*.**
(DOC)

**S1 Fig. Decision strategy for trials independent screening.**
(DOCX)

**S1 File. Search strategy for Pubmed.**
(DOCX)

**S2 File. Extraction form.**
(DOCX)

**S3 File. Results, numerical values for each included outcome.**
(DOCX)

## Acknowledgments

The authors wish to thank Suzanne Rankin for proofreading this manuscript.

## Author Contributions

**Conceptualization:** Anastasia Demina, Benjamin Petit, Vincent Meille, Benoit Trojak.

**Methodology:** Anastasia Demina, Benjamin Petit, Florent Lebon, Benoit Trojak.

**Project administration:** Anastasia Demina.

**Supervision:** Benoit Trojak.

**Validation:** Florent Lebon, Benoit Trojak.

**Writing – original draft:** Anastasia Demina.

**Writing – review & editing:** Vincent Meille, Florent Lebon, Benoit Trojak.

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
