## [Decision Letter · Decision Letter 0]

15 May 2023

PONE-D-23-10930

Interventions combining mindfulness training with non-invasive brain stimulation and their impact on mental health outcomes: protocol for a systematic review and meta-analysis of randomized controlled trials

PLOS ONE

Dear Dr. DEMINA,

Thank you for submitting your manuscript to PLOS ONE. After careful consideration, we feel that it has merit but does not fully meet PLOS ONE’s publication criteria as it currently stands. Therefore, we invite you to submit a revised version of the manuscript that addresses the points raised during the review process.

We look forward to receiving your revised manuscript.

Kind regards,

Carmen Concerto

Academic Editor

PLOS ONE

Journal Requirements:

Reviewers' comments:

Reviewer's Responses to Questions

**Comments to the Author**

1. Does the manuscript provide a valid rationale for the proposed study, with clearly identified and justified research questions?

Reviewer #1: Yes

Reviewer #2: Yes

2. Is the protocol technically sound and planned in a manner that will lead to a meaningful outcome and allow testing the stated hypotheses?

Reviewer #1: Yes

Reviewer #2: Yes

3. Is the methodology feasible and described in sufficient detail to allow the work to be replicable?

Reviewer #1: Yes

Reviewer #2: Yes

4. Have the authors described where all data underlying the findings will be made available when the study is complete?

Reviewer #1: Yes

Reviewer #2: Yes

5. Is the manuscript presented in an intelligible fashion and written in standard English?

Reviewer #1: Yes

Reviewer #2: Yes

6. Review Comments to the Author

You may also provide optional suggestions and comments to authors that they might find helpful in planning their study.

Reviewer #1: This systematic review and meta-analysis protocol for evaluating the effectiveness of a combination of Mindfulness-based intervention and Non-invasive brain stimulation makes a bold attempt to shed light on a highly innovative topic.

Overall, the protocol is well-structured and provides a clear overview of the study aims and methods. All the PRISMA-P points are respected, as authors declared, although it is not clear who the guarantor of study is (item 3b, please specify it in the final draft). The protocol appears to be well-designed and has a comprehensive strategy of how to collect data and manage typical biases and meta-biases which could undermine the results; The authors have thoughtfully considered potential sources of heterogeneity in the data. It also included a valid assessment of certainty of evidence (GRADE).

The authors have also outlined their dissemination plan, which is important for transparency and ensuring that the results of the study reach a wide audience, prudently adhering to the practice of submitting comprehensive reports on the protocols and procedures being implemented during the ongoing study.

Also, I found it particularly notable that this systematic review was conducted with a future original experimental trial in mind, as this approach should always be the norm for this type of research.

However, I would like to point out some minor problems:

□ The abstract lacks information on eligibility criteria and sources of information (databases), which is customary to include. Hence, a more detailed version of the abstract is recommended.

□ The abstract lacks information on eligibility criteria and sources of information (databases), which is customary to include. Hence, a more detailed version of the abstract is recommended.You should also specify the assessment tool which will be used, (RoB 2.0, funnel plot, GRADE etc…) in the abstract, as advisable.

□ “Outcomes” section (line 136) is underdeveloped. I suggest to move here the “Outcomes” section reported later (line 196). Also, the post-COVID mental health status explanation and WHO definition of mental health are not necessary.

□ It is commendable to incorporate studies involving healthy individuals alongside psychiatric patients, as stated in the protocol, provided that the data analyses are conducted separately. It is imperative that this separation is maintained.

□ The inclusion of studies in both clinical and laboratory settings is a good approach as it allows for a comprehensive assessment of the evidence. However, it may be worth considering whether studies conducted in certain settings (e.g. tertiary care centers) are more likely to be included and whether this could potentially impact the generalizability of the findings.

□ It remains unclear whether separate analyses for different interventions and controls will be conducted and presented, in parallel with a pooled pairwise meta-analysis, which is typically recommended. Therefore, please provide clarification on this matter.

□ To provide a more comprehensive and nuanced analysis of your data, I would recommend dividing your analyses into the following time points: less than six months, between six and twelve months, greater than twelve months, and endpoint of the study. By utilizing this standardized subdivision, you can gain a more detailed and accurate understanding of your data within each distinct time period.

□ It is recommended to provide your strategy on how to handle instances where a study presents multiple psychometric scales for a single outcome.

□ To promote transparency and credibility in the studies being analyzed, it is highly recommended to explicitly indicate whether any potential conflicts of interests exist when reporting the characteristics of the studies.

The text is written in good and clear English, with an adequate technical level of expression. Still, there are some minor improvements that can be made. I propose the following suggestions about the writing forms:

□ In line 49, "his/her" can be replaced by the more elegant "their" to avoid gender bias.

□ In line 69, "NIBS are neuromodulation techniques acting focally on the brain surface of specifically predefined cortical areas, as well as on the deeper level" can be rephrased to improve clarity. For example, "NIBS refers to neuromodulation techniques that focus on specific cortical areas of the brain, both on the surface and deeper levels."

□ In line 91, "quantitative syntheses of available data is" should be "quantitative syntheses of available data are" to match the subject-verb agreement.

□ Line 197-201 are pleonastic; I suggest deleting them. Same for lines 206-208 and 220-221.

In light of my previous statements, I recommend accepting this protocol for publication, after some minor revisions.

Reviewer #2: Dear authors,

I am glad to be able to express my views on this protocol for conducting a systematic review with meta-analysis on the combined intervention of mindfulness and non-invasive brain stimulations. I believe the work proposed is of significant clinical interest, given the lack of published aggregate data in the literature.

Regarding the Methodology section, I have not found any particular critical issues. The elements of the PICO are well defined and appear appropriate for the purpose of the work. All aspects relating to statistical analysis (model of statistical analysis to be used, methods for investigating heterogeneity, assessment of the risk of bias) appear correct and justifiable from a methodological point of view.

However, I have some doubts and aspects related to your work.

Organizations of sections

1) I believe it would be appropriate to separate the paragraph on the rationale from the one on the objectives in the introduction section, as suggested by PRISMA-P. From reading the manuscript, the division seems to correspond to line 93.

2) I think that the paragraph "Outcomes and prioritization" (lines 196-255) should be integrated with "Outcomes" (lines 136-140).

Others

- Line 69-70: Here, as in several other parts of the text that I am about to comment on, it is argued that several studies have suggested combining MBI with NIBS. However, there are no references reported here in this regard. I believe it is very important to add adequate references to support these statements as they are extremely important for the background of the work.

- Line 85-87: Again, this statement, although of primary importance in relation to the background of the article, is devoid of any citation.

- Line 273-274: I think it would be appropriate to more thoroughly describe the methods you intend to employ to achieve consensus regarding the trials and the outcomes to be included in the meta-analysis.

7. PLOS authors have the option to publish the peer review history of their article (what does this mean?). If published, this will include your full peer review and any attached files.

Reviewer #1: **Yes: **Pierfelice Cutrufelli

Reviewer #2: **Yes: **Antonio Di Francesco

---

## [Author Response · Author response to Decision Letter 0]

20 Jun 2023

We wish to thank you and the reviewers for their generous comments on our paper. 

We addressed all concerns about PLOS ONE's style requirements and we added captions for our Supporting Information files. We confirm that we did not cite any retracted paper. 

We also edited our manuscript to address all the concerns pointed out by our reviewers.

First reviewer’s concerns:

o We specified the guarantor of the study (L417)

o We added requested details to our Abstract (eligibility criteria and sources as well as software details) (L26-28, L31-33)

o We merged Outcomes and Outcomes and Prioritization sections as requested (L130-177)

o The post-COVID mental health status explanation and WHO definition of mental health were removed.

o We insisted on the separation of clinical and lab studies analyses in our final review (L113-114)

o In our final review we will discuss the issues of generalizability of the findings related to the study settings, among other factors. 

o We planned different subgroup analyses in parallel with a pooled pairwise meta-analysis (L287-290)

o We provided different endpoints for a more comprehensive and nuanced analysis of our data (L263-265)

o We provided our strategy in case of multiple psychometric scales for a single outcome (L274-276)

o We added strategy for handling potential conflicts of interest (L215-216), results will be discussed considering this information.

o All requested changes (L49, L69, L91, L197-201, 206-208, 220-221) were made.

Second reviewer’s concerns:

o Rationale and objectives sections were separated as requested (L94-95)

o We merged Outcomes and Outcomes and Prioritization sections as requested (L130-177)

o We added requested references to support statements pointed out by the second reviewer (L79-80)

o We added our strategy to achieve consensus on studies included in pooled pairwise meta-analysis (L274-276)

---

## [Editor Report · Decision Letter 1]

4 Jul 2023

Interventions combining mindfulness training with non-invasive brain stimulation and their impact on mental health outcomes: protocol for a systematic review and meta-analysis of randomized controlled trials

PONE-D-23-10930R1

Dear Dr. Demina,

We’re pleased to inform you that your manuscript has been judged scientifically suitable for publication and will be formally accepted for publication once it meets all outstanding technical requirements.

Kind regards,

Carmen Concerto

Academic Editor

PLOS ONE
---

## [Editor Report · Acceptance letter]

14 Jul 2023

PONE-D-23-10930R1 

Interventions combining mindfulness training with non-invasive brain stimulation and their impact on mental health outcomes: protocol for a systematic review and meta-analysis of randomized controlled trials 

Dear Dr. Demina:

I'm pleased to inform you that your manuscript has been deemed suitable for publication in PLOS ONE. Congratulations! Your manuscript is now with our production department. 

Kind regards, 

on behalf of

Dr. Carmen Concerto 

Academic Editor

PLOS ONE